# Peer Support for Caregivers of People Living with Posterior Cortical Atrophy in Melbourne, Australia: A Feasibility Study

**DOI:** 10.3390/ijerph21040513

**Published:** 2024-04-22

**Authors:** Alexander Mitchell, Wendy Kelso, Camille Paynter, Leanne Hayes, Dennis Velakoulis, Samantha M. Loi

**Affiliations:** 1Neuropsychiatry Centre, Royal Melbourne Hospital, 300 Grattan Street, Parkville 3050, Australia; alexander.mitchell6309@gmail.com (A.M.); wendy.kelso@mh.org.au (W.K.); leanne.hayes@mh.org.au (L.H.); dennis.velakoulis@mh.org.au (D.V.); 2Melbourne School of Health Sciences, University of Melbourne, Parkville 3052, Australia; camille.paynter@unimelb.edu.au; 3Department of Psychiatry, University of Melbourne, Parkville 3052, Australia

**Keywords:** young-onset dementia, posterior cortical atrophy, peer support group, carer support group

## Abstract

Posterior Cortical Atrophy (PCA) is a rare form of young-onset dementia that causes early visuospatial and visuoperceptual deficits. The symptom profile of Posterior Cortical Atrophy leads to very specific care needs for those affected, who often rely on informal caregivers (including friends and family). Rare dementia support groups can be useful for both patients and their caregivers to assist with knowledge sharing, psychoeducation, and the provision of psychosocial support. Despite this, few such support groups exist. The purpose of this study was to examine a PCA support group for caregivers of individuals living with PCA. We held a structured psychoeducation support group comprised of four sessions with the aim being to provide education, strategies for the management of the disease, and peer support. Caregivers’ mental health and quality of life were assessed. The results of our study showed that support group participation was a positive experience and assisted with increasing the knowledge of caregivers and fostering social connections. We suggest that peer support groups may be beneficial for both people living with PCA and their caregivers. We recommend that future quantitative and qualitative research is conducted to further assess health-promotion benefits to people living with PCA and their caregivers, and to assess their development and implementation in different contexts.

## 1. Introduction

Posterior Cortical Atrophy (PCA) is a rare form of young-onset dementia (YOD), and the age of onset is generally between 50 and 65 years of age [1]. ‘Posterior Cortical Atrophy’ was first introduced by Benson and colleagues after observing a group of individuals with progressive dementia, sharing early higher-order visual processing dysfunction and the development of alexia, agraphia, visual agnosia, and components of Balint’s, Gerstmann’s, and transcortical sensory aphasia syndromes [2]. The visual disturbances in PCA are not due to ophthalmological or ocular pathology [2,3]. Memory, insight, behaviour, and personality in PCA are relatively preserved early in the course of the disease, unlike in Alzheimer’s dementia (AD) [1,2]. PCA is most commonly associated with pathology related to AD; however, other rare associations include Lewy body disease, corticobasal degeneration (CBD), and prion disease [3,4]. In the natural history of PCA, the earliest clinical findings include the aforementioned visuospatial and visuoperceptual deficits, followed later by deficits in language, executive function, and memory, as well as progression to functional blindness [1,5]. Diagnosis of PCA involves neurological and neuropsychological evaluation, as well as structural brain imaging, usually via magnetic resonance imaging (MRI). Functional imaging such as fluorodeoxyglucose positron emission tomography (FDG-PET) and single-photon emission computed tomography (SPECT) may be utilised if further imaging is required [5]. Cerebrospinal fluid confirming the presence of AD pathology, low amyloid-β (Aβ1-42), and raised total- and phosphorylated-tau or plasma biomarkers may be useful in both ruling out alternative causes of symptoms or confirming the underlying pathological process driving PCA [5]. The most recent international consortium criteria include clinical features (insidious onset, gradual progression, early disturbance of visual with or without posterior cognitive features); at least three cognitive features (such as Gerstmann’s syndrome—acalculia, right–left disorientation, finger agnosia, agraphia, apraxia, alexia and simultanagnosia); and relatively spared memory function, speech and non-visual language function, executive function, and behaviour and personality, with predominant occipitoparietal or occipitotemporal atrophy or hypometabolism on MRI/FDG-PET/SPECT [6]. 

The management goals of all dementias, with no curative treatments available, include implementing strategies to enhance functional independence while managing symptoms [7]. Pharmacological approaches can be used but are restricted to symptomatic treatment, with some dementias, like AD, having limited options available, while other dementias have none [7]. Non-pharmacological approaches include multidisciplinary team involvement, caregiver education, environmental and lifestyle modifications, behaviour management, and psychosocial support groups [7]. Support groups specifically are an important part of disease management, as they are of significant benefit for increasing the quality of life in individuals with dementia and their caregivers [5,8]. Despite some overlap in symptoms and underlying pathological processes, it is recognised that those with YOD are distinct from older people with dementia in terms of their care and support needs [9,10]. Individuals with YOD have unique psychosocial challenges in terms of their relationships (e.g., they may have a family with young children), employment and finances (e.g., they may have been working to support themselves or their family), and support (e.g., they patients may need to navigate a health, aged, or disability care service that may not be set up for younger people) [8,11,12]. Services do not often cater to the specific practical or psychological needs of younger people with dementia, which is a barrier for those with YOD accessing these services [9,10,13]. Support services that are focused on the specific needs of people with YOD have been reported as positive [9,10]. Consistent with this, considering the broad aetiologies of YOD, support groups for YOD should be tailored to the symptoms profile, care needs and concerns of the specific type of YOD. For example, in behavioural-variant frontotemporal dementia, a YOD characterised by behaviour and personality change and executive impairment, support groups have focused on positive behaviour support for caregivers for behaviours such as apathy, irritability, and disinhibition [14,15].

This is exemplified well for people with PCA who have specific needs and care considerations based on the symptom profile. The progression of visuospatial impairment in PCA leads to reduced participation in activities of daily living and higher rates of disability relatively early in the disease process, requiring specific care for these individuals from their caregivers [5,16,17]. PCA is a rare type of dementia and is not well known even among the dementia community. This lack of available information has also been reported as a significant cause of stress for those living with PCA and their caregivers [18]. Peer support groups are vital to provide pathways to connect with others with the condition and offer a positive impact by increasing knowledge and understanding of the condition, and by fostering social connection with peers to reduce feelings of isolation [5]. Despite the reported benefits of assisting people with PCA in navigating these specific challenges and improving their quality of life, very few support groups specifically for those living with PCA have been developed. For example, Rare Dementia Support offers PCA meetings to provide education and discuss strategies and support for young people living with PCA [19].

Caregivers of individuals with YOD experience high levels of burden; depression and anxiety; psychosocial challenges including interpersonal difficulties; financial and occupational concerns; and burnout [12,20,21]. The causes of these are multifactorial but may be due to the low availability of specific support services and resources available to assist them with coping and managing in their caregiver role [10,21]. Individuals with YOD are more likely to receive care in the community for more than twice as long as those with older-onset dementia, which may contribute to heightened caregiver distress and burnout [11,22]. Caregiver burden is associated with poorer outcomes not only for the caregiver but also for the person with dementia, and this may lead to earlier institutionalisation [21,22]. Caregiver support groups that are YOD-specific can be beneficial in providing education and facilitating coping, which may reduce psychosocial symptoms such as anxiety and burden [20,23].

PCA is a debilitating illness for young people, often leading them to require significant support from caregivers. Despite this, there is minimal literature reporting on support groups for PCA, particularly in Australia. There has been one evaluation of a PCA support group, which comprised four sessions delivered in Melbourne, Australia, in 2015/2016 in conjunction with Dementia Australia (previously Alzheimer’s Australia) [13]. This group included both caregivers and people with PCA; provided education, addressed specific areas of concern; and allowed time for discussion and connection between caregivers and people with PCA as well as the sharing and development of resources, skills, and strategies. In brief, the four sessions focused on topics titled ‘Awareness of PCA’, ‘Tips for enhancing cognition and function’, ‘Language and communication’ and ‘Visual issues that are not about the eyes’. The evaluation reported that all participants increased their knowledge after each session and found the facilitators to be insightful and knowledgeable, and that friendships and support networks improved following the sessions. There were several challenges acknowledged. The group had an ‘open’ selection criteria format, which meant that there was a range of participants in attendance, at differing stages of disease progression. This may have been confronting for those with mild or early symptoms, and it was difficult to tailor discussions to the wide range of needs. Because there had not been a PCA support group run previously, the demand was high, and the group was deemed too large. The facilitators reported challenges in ‘containing the uncontained’ while ‘supporting the silent voice’. Practically, keeping to time was difficult due to high group numbers; using a PowerPoint presentation was not useful for those with PCA (due to their visual impairment); and it was acknowledged that issues such as the referral source for entering and exiting the group and the lack of inclusion/exclusion criteria needed to be addressed in future.

Building on the learnings of the original 2015/2016 group’s evaluation, we report on a PCA support group for caregivers whose family members were in the early stages of the diagnosis. The aims of the group were to improve knowledge of PCA, provide strategies for managing PCA, and importantly, provide a support network for caregivers looking after individuals with rare dementia.

## 2. Materials and Methods

### 2.1. Study Design

This was a non-blinded feasibility study of a structured psychoeducation support group, primarily for informal caregivers of people living with PCA. The individuals with PCA for whom they were providing care were invited to attend but were not the focus of the intervention. The intervention was comprised of four sessions developed from the evaluation report from the previous Dementia Australia PCA support group. Four 2-hourly caregiver psychoeducation support sessions were designed using previous feedback on topics that had been reported as useful by evaluation participants and in consultation with individuals living with PCA and their care partners. (Table 1).

Each session was structured so that the first hour had one topic followed by a 20 min morning tea break, and then there was a second one-hour session. In the second half of the session, the people with PCA and caregivers were split (if feasible) so they could have a session on their own with one of the facilitators.

### 2.2. Participants and Recruitment

Potential participants were caregivers whose family members with PCA attended Royal Melbourne Hospital Neuropsychiatry Centre and Dementia Australia. Inclusion criteria for the caregivers included: 1. Consider themselves a caregiver for someone with PCA; 2. Be able to attend four fortnightly, face-to-face sessions; 3. The individual with whom they were caring for was diagnosed with PCA in the previous 5 years; 4. Have sufficient English language to understand and participate during the sessions; and 5. Have adequate hearing and vision to participate during the sessions. A screening interview was conducted with the caregiver and person with PCA prior to group participation, to ensure they met the inclusion criteria and were suitable for group inclusion. This introductory interview provided information about the proposed group, with an opportunity to clarify expectations and answer questions. The caregiver was able to attend with the person they were supporting, or alone, depending on their preference.

### 2.3. Questionnaires

While this group was not primarily focused on improving the mental health of caregivers, pre-group measures of the caregivers and people with PCA were completed to provide a quantification of their mental health and other parameters. Scales administered included the Depression Anxiety Stress Scale (DASS-21), the Zarit Burden Interview, and the EQ5D. The DASS-21 has three subscales measuring depressive, anxiety, and stress symptoms on a four-point Likert scale (0 not at all to 3 most of the time), with scores multiplied by 2 [24]. Higher scores indicate worse mental health symptoms. The Zarit Burden Interview contains 22 statements of burden with a five-point Likert scale (0 never to 4 nearly always), with higher scores indicating increased levels of burden [25]. The EQ5D is a general measure of health asking respondents to indicate how much of a problem certain activities are, such as mobility, self-care, and pain, on a five-point Likert scale (1 no problem to 5 unable to/extreme), and it also incorporates a visual analogue scale (VAS) for rating quality of health from 0 (worst) to 100 (best) [26]. These measures all have adequate reliability and validity.

An evaluation of the group was completed by the caregivers using a questionnaire following the final session, which measured caregivers’ perceptions of the relevance of the content, knowledge gained, the ease of following the facilitator/s and overall experience of the sessions, with a free text box for additional comments. This evaluation was based on a similar framework used to evaluate the 2015/2016 PCA support group [13].

Ethics approval was provided by the Royal Melbourne Hospital Human Research Ethics Committee (approval ID: #QA2020004).

## 3. Results

This study took place at a Dementia Australia site located in metropolitan Melbourne, with four fortnightly sessions occurring from 27 August 2019 to 7 October 2019. There were five caregivers who participated, and the individuals living with PCA, for whom they provided care, also attended. Of the caregivers, two were referred by private specialists—one from the Royal Melbourne Hospital (RMH) and one from Dementia Australia. There were four facilitators present at all sessions—a staff member from Dementia Australia, a neuropsychiatrist, a neuropsychologist, and a social worker from the RMH.

Demographics and pre-group health measures of the caregiver participants and those that they were caring for are shown in Table 2 and Table 3. In general, caregivers had an average age of 66 years old and nominated that they were in the early years of providing care for people with PCA who had mild (three, 60%) and moderate (two, 40%) disease severity. They were all spousal caregivers. In terms of their mental health, all had mild depressive, anxiety, and stress symptoms. Caregiver burden was in the mild-to-moderate range, with quality of health rated as very good. All but one caregiver described having minimal problems with their health. One caregiver who rated ‘extreme’ on the EQ5D and reported severe levels of depression, anxiety, and stress was referred to the psychiatrist on the team, who performed a risk assessment, and they were referred to see their general practitioner.

People with PCA had moderate symptoms of depression, anxiety, and stress; lower self-rated quality of health compared to caregivers; and indicated severe problems in various aspects of health such as usual activities and personal care. Three (60%) reported that their initial symptoms were visual-related, such as ‘eyesight problems’ and ‘visual field problems’.

On average, four (80%) caregivers and their spouses with PCA attended the four sessions. The evaluation following the final session was completed by four caregivers (80%), who reported increased knowledge of PCA, particularly with regard to managing the cognitive symptoms (Table 4 and Table 5). ‘Free text’ comments from the caregivers included statements such as ‘group discussion helpful, hearing problems others face are similar to ours’, ‘hearing from other PCA folks directly from their experience’, ‘meeting everyone, feeling supported, loved being here, part of the group’, ‘hearing from others who had the same disease’, and ‘helpful hearing each others’ issues, sharing experiences’. Of note, caregivers found the second half of the session (when the people with PCA had their own group session) very useful. Participants also added topics they wished to learn more about, including ‘other support groups’, ‘how to tell others (family and friends) about PCA’ and ‘almost anything, we are hungry to learn and become educated’.

## 4. Discussion

This paper aimed to report on a support group for the informal caregivers of people with PCA and their family members living with PCA. The support group was designed with a focus on supporting informal caregivers and increasing their knowledge of PCA, as well as providing a support network. We expanded on the format and addressed the limitations of the PCA support groups run in 2015/2016 [13]. This included conducting a pre-group attendance screening interview, developing clearer inclusion and exclusion criteria, adhering to a ‘closed’ group structure with a small number of caregivers, restricting the group to those with early-stage PCA, and having a limit of four sessions to provide only relevant information and practical skills. In general, the evaluation showed that the sessions were useful in terms of learning more about PCA; how to manage the visual, cognitive, and behavioural symptoms; and how to access more resources. They also provided an opportunity to meet others in the same situation.

The participants of our group were not dissimilar to caregivers and people living with PCA described in the literature [17,18]. Caregivers in our study nominated mostly low levels of anxiety and depression, with mild to moderate caregiver burden. However, one caregiver did nominate severe anxiety and depression. While these results suggest that these caregivers appeared to be managing relatively well, this could be biased as the participant caregivers were at an early stage of providing care. It can be hypothesised that caregivers of people living with PCA with more pronounced symptoms may show higher levels of psychosocial symptoms and burden, due to the type and increased level of care required. In contrast, the individuals living with PCA (all provided baseline data) reported high levels of anxiety, stress, and depression on self-report measures. Previous studies have reported that people living with PCA have more mental health concerns compared to other types of dementia—for example, anxiety was more prominent in those with PCA compared to those with usual Alzheimer’s disease [27]. Depression may also be more prominent in those with PCA, possibly due to relatively preserved insight in the early stages of the disease [1]. This is understandable given the burden of their illness and symptom profile (e.g., visuospatial deficits) and the impact of this on their function. This was reflected in the baseline EQ5D function responses from more than half the participants with PCA, who reported ‘a moderate problem’ in all functional domains. Of note, 80% of the people with PCA reported ‘Moderate’ or ‘Severe’ problems with self-care, and 60% reported ‘Severe’ problems with participation in usual activities, despite having early-stage disease. Significant impairment in self-care and everyday tasks has also been noted previously in the PCA literature, with people becoming functionally impaired at an earlier disease stage than in other dementias due to cortical blindness and visuospatial and visuoperceptual difficulties [17]. People living with PCA often have difficulties describing the exact nature of their unusual visual and cognitive changes, which can heighten the sense of isolation. As PCA is considered rare, there is a lack of understanding of this specific type of dementia, including among healthcare professionals. This limited disease-specific knowledge and understanding often leads to a delay in diagnosis, with the person feeling dismissed, misunderstood, and lonely [28,29]. These experiences can heighten levels of anxiety, as the person with PCA knows ‘something is wrong’; however, their symptoms differ significantly from commonly known dementia syndromes. This illustrates why specific support groups are essential for caregivers of people living with PCA, as there are unique challenges for care provision due to the rare symptom profile and level of functional impairment.

Positive aspects of the group were reported in the ‘free text’ evaluation questions and related to the second hour of each session, in which people living with PCA and caregivers were separated into different groups. This allowed each group the opportunity to discuss their specific concerns, raise questions and relate experiences with each other, in the presence of a group facilitator. The group division was well-received by all participants, allowing them time to address their specific needs and concerns in a safe and confidential space, without the presence of their care partner/spouse, or the person they were caring for. Furthermore, responses such as ‘sharing experience’ and ‘meeting more people in my situation’ highlight how participants found these groups useful for facilitating social connection and support.

It should be noted that the support group was run in person, as opposed to online through telehealth, which is both a strength and a limitation. In-person support groups have been shown to be beneficial in fostering the formation of friendships and a support network and allow for physical connection and support, which was noted to be missing by some participants using online-based services [13,30]. In-person support groups are also beneficial, as participants may not feel like they are speaking to a real person if participating in an online group format [31,32]. Furthermore, there may be technological challenges or barriers, such as accessing the internet and navigating technology, particularly for those who are visually impaired, often making in-person support groups preferable [31,33]. Alternatively, online support groups can also be helpful, as they may be more accessible for people with dementia and their caregivers, who may feel more comfortable and less vulnerable in their home setting [30]. Caregivers commonly report significant challenges attending in-person caregiver support groups due to the difficulties associated with organising respite care for the person with dementia. Additionally, people in remote locations have the ability to access and engage with support groups if they are delivered online [32]. Subsequent to the COVID-19 pandemic restrictions, and the rarity of a YOD such as PCA, ongoing viability for specific PCA support groups may mean that an online format is required. A recent online music therapy and cognitive behaviour therapy intervention for people with YOD and their caregivers cited benefits from the online format [34] and we would recommend trialling an online version with a larger sample size, to investigate the acceptability and feasibility of this setup.

The small sample size of the group was a limitation and possibly meant there was a biased selection. There were only five caregivers involved, with 80% completing the four sessions and completing the evaluation. We had positive results from the evaluation which included responses from participants about topics they wish to learn more about. These included ‘how to tell people (family and friends) about PCA’ and ‘almost anything, we are hungry to learn and become educated’. This information demonstrates a need to improve community awareness of this rare type of dementia and the need for further knowledge and information.

Future research may consider in-person and online support groups of a larger size with open recruitment to assess the feasibility and acceptability of these groups, with the ultimate goal being progression to the establishment of regular support services for people living with PCA and their caregivers. The investigation of an ongoing support group with a less-structured, more peer-focused approach may yield meaningful information about the impact of these groups on rates of anxiety, stress, depression, and burden. Consideration of how to provide care for the person with PCA so that their caregiver can ‘attend’ the groups is also required. Co-designing the format of these groups would be an essential step for the development of these. Furthermore, understanding the psychosocial profile of caregivers is important in designing support groups, and thus more research into depression, anxiety, burden, grief, and loss at all stages of the caregiving journey would be useful. Designing a support group specifically for people with PCA (rather than caregivers) is also an important gap in the field.

The results of this feasibility study suggest there are discernible benefits for caregivers of people living with PCA. We recommend that online and in-person support groups continue to be developed and delivered across Australia in rural and metropolitan settings. The ongoing evaluation of developing support groups will further contribute to service delivery research in Australia and internationally.

## 5. Conclusions

PCA is a rare form of YOD characterised by visuospatial and visuoperceptual deficits that gives rise to significant functional impairment in mid-life. Due to the debilitating nature of the condition, tailored individual support is required early on in the disease from informal caregivers. Few specific support services for those with PCA and their caregivers exist to provide education and assist with managing day-to-day function and psychosocial symptoms. Learning from a previous group, we designed and ran a small-scale peer support group for caregivers supporting individuals living with PCA, which was well received, and provided some benefits in terms of increasing knowledge and fostering social connections. We suggest that further support groups be developed and implemented to support caregivers and people living with PCA and that these support groups have co-designed evaluation and research components. Co-designing support groups including their evaluation allows for greater participant involvement in program development and enables them to become advocates for the type of support they need.

## Figures and Tables

**Table 1 ijerph-21-00513-t001:** Session Outline.

Session	Topic/Theme	Speakers
Session 1	Introduction—What is PCA? (including information regarding visual changes)	Medical, Neuropsychologist, Social Worker
Session 2	Support, Genes and Treatments	Medical, Neuropsychologist, Social Worker
Session 3	Strategies and Quality of Life	Occupational Therapist
Session 4	Speech and Communication	Speech Pathologist

PCA = Posterior Cortical Atrophy.

**Table 2 ijerph-21-00513-t002:** Participants’ demographics and pre-group health measures.

	Mean (SD)	Median (IQR)	Range
Caregivers (N = 5)			
Women, N (%)	2 (40%)		
Age in Years	65.8 (9.6)	68 (56, 74.5)	53, 76
Duration of Caring in Years	3.0 (1.7)	3.5 (1.3, 4.5)	0.5, 5
DASS21-Depression	10.2 (15.0)	12 (7.0, 24)	0, 36
DASS21-Anxiety	8.8 (18.4)	14 (2, 22)	0, 40
DASSS21, Stress	15.2 (15.2)	14 (10, 20)	6, 42
ZBI	37.2 (12.1)	42 (40.5, 47)	19, 52
EQ5D VAS	72.5 (29.8)	80 (55, 90)	30, 100
Person with PCA (N = 5)			
Women, N (%)	3 (60%)		
Age in Years	65.8 (8.6)	67 (57, 74)	56, 75
DASS21-Depression	17.2 (14.8)	28 (24, 30)	0, 34
DASS21-Anxiety	12.4 (12.2)	14 (24, 30)	0, 34
DASS21-Stress	19.6 (10.6)	18 (17, 28)	6, 38
EQ5D VAS	63.7 (12.7)	66 (58, 70.5)	50, 75

DASS21: Depression Anxiety Stress Scale; ZBI: Zarit Burden Interview; PCA: Posterior Cortical Atrophy; VAS: Visual Analogue Scale.

**Table 3 ijerph-21-00513-t003:** Participants’ EQ5D Function.

	No Problem	Slight Problem	Moderate Problem	Severe Problem	Unable/Extreme
Caregivers (N = 5)					
Mobility	5 (100%)				
Personal Care	5 (100%)				
Usual Activities	5 (100%)				
Pain/Discomfort	4 (80%)	1 (20%)			
Anxiety/Depression	3 (60%)	1 (20%)			1 (20%)
Person with PCA (N = 5)					
Mobility	2 (40%)	1 (20%)	2 (40%)		
Personal Care	1 (20%)		3 (60%)	1 (20%)	
Usual Activities		2 (40%)		3 (60%)	
Pain/Discomfort	2 (40%)	1 (20%)	1 (20%)	1 (20%)	
Anxiety/Depression		1 (20%)	2 (40%)	2 (40%)	

PCA: posterior cortical atrophy.

**Table 4 ijerph-21-00513-t004:** Caregiver participant evaluation of the PCA support group ‘Knowledge about PCA’.

Question	Strongly Disagree	Disagree	Neither	Agree	Strongly Agree
I have learnt more about PCA					4 (100%)
I know more about how to manage the visual symptoms				1 (25%)	3 (75%)
I know more about how to manage the cognitive symptoms (e.g., memory and language deficits)					4 (100%)
I know more about the ways to manage the behavioural symptoms (e.g., agitation and wandering)				2 (50%)	2 (50%)
I know more about how to manage my carer stress and burden				2 (50%)	2 (50%)
I know more about where to access further help and resources				1 (25%)	3 (75%)
I have met others who are caring for people with PCA					4 (100%)

PCA: Posterior Cortical Atrophy.

**Table 5 ijerph-21-00513-t005:** Caregiver participant evaluation of the PCA support group.

Question	Strongly Disagree	Disagree	Neither	Agree	Strongly Agree
The sessions met my expectations					4 (100%)
The topics in the sessions were relevant to me					4 (100%)
There was time for questions and discussion					4 (100%)
Participating in the sessions have helped me in my caring role					4 (100%)
The duration of sessions (2 h) was sufficient					4 (100%)
I felt more knowledgeable about speech and communication					4 (100%)
The facilitators were easy to follow					4 (100%)

## Data Availability

The original contributions presented in the study are included in the article, further inquiries can be directed to the corresponding author.

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
