# Peer review of "Peer Support for Caregivers of People Living with Posterior Cortical Atrophy in Melbourne, Australia: A Feasibility Study"

_ijerph, 2024, doi:10.3390/ijerph21040513_

Round 1

Reviewer 1 Report

Comments and Suggestions for Authors

I would recommend some reference to the methods (e.g., the questionnaire and analysis) are included in the abstract.

Cronbach alpha's for the included quantitative scales should be reported. 

There is some information missing from the results section. For example, in the discussion the following is mentioned and discussed: "“free text” evaluation questions 276 and related to the second hour of each session". I would recommend the discussion of what was reported by participants in these free-text elements should be included in the results section. 

Author Response

Thank you very much for these important comments. 

As suggested we have included selected information about the Methods in the Abstract “Caregiver mental health and quality of life was assessed”.

Cronbach’s alpha coefficient measures the internal consistency or reliability when designing and testing a new survey.  We did not include Cronbach’s alpha on our evaluation questionnaire as the purpose of the evaluation was to assess how the caregivers found the support group, rather than a reviewing whether our evaluation questionnaire would be a new assessment instrument.

There is some information missing from the Results section.  We have reviewed this and ensured that information from the Discussion is first mentioned in the Results, including the “Free text” information.  In the Methods (line 184) and Results (line 219) there is mention of the “free text” box.

Reviewer 2 Report

Comments and Suggestions for Authors

Comments:

This is a study assessing the peer support in posterior cortical atrophy for patients and their care givers in Melbourne. The authors report a positive experience and increased knowledge in caregivers. The research question is relevant form the health care perspective. However, there are several major limitations in the study methodology. The study is more of a survey. The manuscript is well structured and easy to read. The authors need to address the following points:

1-       The introduction and justification of the study are appropriate. Authors should highlight how the peer support for care givers will be different for PCA in comparison to other types of dementia.

2-       The major limitation of the study is the sample size of 5. Any scientific conclusion from such a limited sample can be erroneous. This limits external validity of the study.

3-       Out of five participants, 4 completed the four sessions and the outcome (participant evaluation) was assessed in these 4 participants. The outcomes with such small sample size are more likely to be skewed and the conclusions thus arrived is likely to have limited generalizability.

4-       It would have been interesting if the investigators had evaluated the caregiver stress after a definite period post intervention.

5-       The satisfaction level of the participants may not be an ideal outcome for such studies. Instead, the post intervention changes in depression, stress level, quality of life of caregiver and patients would have been more clinically useful outcomes.

6-       The change in knowledge would have been evaluated after a certain period. Post intervention outlook, perspectives about the disease would have been more useful.

Comments on the Quality of English Language

Minor editing of the English language is required. 

Author Response

RESPONSE TO REVIEWER 2

We thank Reviewer 2 for their valuable comments and have responded to these.

The introduction and justification of the study are appropriate. Authors should highlight how the peer support for care givers will be different for PCA in comparison to other types of dementia.   Thank you, this is an important comment which we address in the Introduction on pg 2 “For example, in behavioural-variant frontotemporal dementia, a YOD characterised by behaviour and personality change and executive impairment, support groups have focused on positive behaviour support for caregivers, for behaviours such as apathy, irritability and disinhibition (O’Connor et al 2019)”

The major limitation of the study is the sample size of 5. Any scientific conclusion from such a limited sample can be erroneous. This limits external validity of the study.  Thank you, we acknowledge that a major limitation from our study was the small sample size. 

Out of five participants, 4 completed the four sessions and the outcome (participant evaluation) was assessed in these 4 participants. The outcomes with such small sample size are more likely to be skewed and the conclusions thus arrived is likely to have limited generalizability.

It would have been interesting if the investigators had evaluated the caregiver stress after a definite period post intervention.  Thank you for this comment and we agree this would have been interesting however longitudinal follow-up was beyond the scope of this study.

The satisfaction level of the participants may not be an ideal outcome for such studies. Instead, the post intervention changes in depression, stress level, quality of life of caregiver and patients would have been more clinically useful outcomes.  Thank you for this response as mentioned in the Methods due to the small sample size, we did not think evaluating the caregivers outcomes following the support program would provide significant results.  However we did wish to evaluate what caregivers thought about the program itself and used an evaluation questionnaire as detailed in the report.

The change in knowledge would have been evaluated after a certain period. Post intervention outlook, perspectives about the disease would have been more useful.  Thank you for this comment and we agree this would have been interesting however longitudinal follow-up was beyond the scope of this study.

Reviewer 3 Report

Comments and Suggestions for Authors

Peer Support in Posterior Cortical Atrophy for Patients and their Caregivers in Melbourne Australia. 

This is a really interesting and encouraging study describing a peer support intervention for caregivers of people living with Posterior Cortical Atrophy (PCA). The argument for PCA-specific support groups is strongly made and the feedback given by the participants points to the importance and need for such groups and the particular success of the authors’ intervention. Reading through, I had concerns over the small number of participants and the pre-COVID issue, but the authors addressed these limitations in the discussion section. The intervention described here lays the foundation for larger-scale (perhaps online/telehealth or hybrid) peer-support group interventions, and the insights are certainly worthy of publication despite the small number of participants. I offer a few comments below that I think will improve the manuscript.

Major comments

Introduction: I think the Introduction section gives a lot of detailed background information and is presented in a logical way; however, for telling the story of your study—which is focused on the “peer support” intervention aspect rather than the disease itself—my suggestion would be to rearrange the paragraphs to begin with a discussion off the need for dementia support groups. For example, you could begin with the current second paragraph, starting: “The management goals…” (Line 58). Then move the description of PCA (currently the first paragraph, lines 28-57) to the current third paragraph (lines 80-94). It would flow very nicely from line 84, where you state again about the rarity of PCA. Further, I think you could give a more concise description of PCA, cutting some of the superfluous information (e.g., the diagnostic criteria) unless you felt it was necessary to your narrative. 

Section 2.1: As a reader, I’d like to know more about the intervention beyond what is given in the Introduction, section 2.1, and Table 1. Is this described in previous publications or available for the reader to reference somewhere? Maybe Reference 13? Perhaps it is outside of the study’s scope, but if isn’t available elsewhere, I think it would be good to offer more details of the intervention’s content, possibly by expanding Table 1 or as an Appendix. 

Section 2.3: I think it would be good to provide more detail on each of the instruments used in the study, in terms of basic cut-off scores (if appropriate) and validity and reliability. 

Minor comments

Section 2.2: I think it would be good to include a statement here about participants giving informed consent for inclusion in the study.

Line 184: Grammatical mistake in the ethics statement: “…was provided by the…”. the word “committee” should also be capitalized.

Lines 188-192: The two sentences here both begin with “There were five caregivers who participated…” I think these sentences should be combined.

For clarity, I advise using the “serial comma” or “Oxford comma” throughout the manuscript. 

Comments on the Quality of English Language

The English is excellent throughout the paper, a few minor issues were detected, which I mention in the comments above.

Author Response

We thank Reviewer 3 for their useful comments.

Major comments

Introduction: I think the Introduction section gives a lot of detailed background information and is presented in a logical way; however, for telling the story of your study—which is focused on the “peer support” intervention aspect rather than the disease itself—my suggestion would be to rearrange the paragraphs to begin with a discussion off the need for dementia support groups. For example, you could begin with the current second paragraph, starting: “The management goals…” (Line 58). Then move the description of PCA (currently the first paragraph, lines 28-57) to the current third paragraph (lines 80-94). It would flow very nicely from line 84, where you state again about the rarity of PCA. Further, I think you could give a more concise description of PCA, cutting some of the superfluous information (e.g., the diagnostic criteria) unless you felt it was necessary to your narrative.  Thank you for these detailed suggestions for reformatting the Introduction, however in lieu of the rapid turnaround for the revisions, we have decided to keep the current order.  However we agree that the description of PCA could be more concise and have reduced the clinical description of PCA as suggested (pg. 3 and 4). We have also amended the Discssion to improve readability. 

Section 2.1: As a reader, I’d like to know more about the intervention beyond what is given in the Introduction, section 2.1, and Table 1. Is this described in previous publications or available for the reader to reference somewhere? Maybe Reference 13? Perhaps it is outside of the study’s scope, but if isn’t available elsewhere, I think it would be good to offer more details of the intervention’s content, possibly by expanding Table 1 or as an Appendix.  We appreciate Reviewer 3’s interest in the intervention beyond what is provided in the manuscript and while the intervention is based on Reference 13 with improvements (as detailed in the Discussion), we have not published the protocol.  The corresponding author can be contacted for more information.

Section 2.3: I think it would be good to provide more detail on each of the instruments used in the study, in terms of basic cut-off scores (if appropriate) and validity and reliability.  Thank you, we have added that these measures have “adequate reliability and validity”.  We chose not to include the cut-off scores for brevity but have the references if readers wish for more details (pg. 5).

Minor comments

Section 2.2: I think it would be good to include a statement here about participants giving informed consent for inclusion in the study.  Thank you, the informed consent statement is at the end of the manuscript.

Line 184: Grammatical mistake in the ethics statement: “…was provided by the…”. the word “committee” should also be capitalized.  Thank you, we have amended.

Lines 188-192: The two sentences here both begin with “There were five caregivers who participated…” I think these sentences should be combined.  Thank you, we have combined these sentences as suggested (pg. 5).

For clarity, I advise using the “serial comma” or “Oxford comma” throughout the manuscript.  Thank you, we have used the “Oxford comma” as suggested.

Round 2

Reviewer 2 Report

Comments and Suggestions for Authors

The explanations by the authors to my previous queries are not satisfactory. 

Author Response

We have made amendments according to the Editors' suggestions